# High-Complexity WO₃-Based Catalyst with Multi-Catalytic Species via 3D Printing

**Xiaofeng Wang** [1] , **Wei Guo** [2] , **Raed Abu-Reziq** [2] **and Shlomo Magdassi** [2,*]

[1]   School of Materials Science and Engineering, Central South University, Changsha 410083, China; wangxiaofeng@csu.edu.cn

[2]   Institute of Chemistry and Casali Center of Applied Chemistry, The Hebrew University of Jerusalem, Jerusalem 91904, Israel; guowei198905@126.com (W.G.); Raed.Abu-Reziq@mail.huji.ac.il (R.A.-R.)

*   Correspondence: magdassi@mail.huji.ac.il; Tel.: +972-2-6584967

**Abstract:** Three-dimensional (3D) printing has recently been introduced into the field of chemistry as an enabling tool employed to perform reactions, but so far, its use has been limited due to material and structural constraints. We have developed a new approach for fabricating 3D catalysts with high-complexity features for chemical reactions via digital light processing printing (DLP). PtO₂-WO₃ heterogeneous catalysts with complex shapes were directly fabricated from a clear solution, composed of photo-curable organic monomers, photoinitiators, and metallic salts. The 3D-printed catalysts were tested for the hydrogenation of alkynes and nitrobenzene, and displayed excellent reactivity in these catalytic transformations. Furthermore, to demonstrate the versatility of this approach and prove the concept of multifunctional reactors, a tungsten oxide-based tube consisting of three orderly sections containing platinum, rhodium, and palladium was 3D printed.

**Keywords:** 3D printing; digital light processing; chemical reaction; catalyst; WO₃; PtO₂; multi-catalytic species

## 1. Introduction

Three-dimensional (3D) printing, also known as Additive Manufacturing (AM), is a bottom-up fabrication technique based on the sequential layer-by-layer deposition of a material, in which physical objects are assembled from digital models. In the past decade, this technique, which was proposed in the mid-1980s, has been rapidly developed: Not only have categories of printing paths based on different basic principles, such as extrusion [1], jetting [2,3], fusion [4–6], stereolithography [7,8], etc. flourished [9], but its innovative application domain has been exploited and extended in several fields, such as biology [10,11], medicine [12,13], electronics [14–16], fabricating complex parts [17,18], and more [19,20]. Therefore, 3D printing is regarded as a vital part of the fourth industrial revolution due to its potential capabilities for fabricating customized and personal products on site, as well as its numerous benefits, including rapid prototyping, a high efficiency, and cost-time saving, etc. [21,22]. Furthermore, 3D printing has recently been introduced into chemistry and chemical engineering to control chemical processes [23] due to its advantages, such as its ability to significantly enhance the feasibility of controlling fluid dynamics during a reaction and validate the benefit of complex 3D shapes with computationally optimized geometries. For example, Leroy Cronin et al. made use of 3D printing to create reaction-ware and thus to adjust the reaction outcome by altering controllable architectures, as well as the composition [24,25], to conduct hydrothermal synthesis with sealed reactors [26]. They demonstrated a strategy in which a piece of custom software is used to control the synthesis parameters via 3D-printed robotic equipment [27]. Additionally, 3D printing has been combined with the advantages of flow chemistry for the synthesis of organic compounds [28,29].

More specifically, 3D printing has been utilized to control catalysis reactions [30], especially for heterogeneous catalytic materials based on ceramic materials [31,32], for which the key properties are highly dependent on the fabrication method (e.g., morphology and dimensions of the system and loading levels of the catalyst), besides the intrinsic features of its catalytic species and solid supports. Catalytic species were previously introduced into 3D matrices by the dispersion of reactive particles into the inks of solid supports [33,34] or by post-printing treatment of 3D-printed solid supports, such as chemical surface modification [35]. However, although, in principle, 3D printing offers the possibility to control chemical reactions with accurate performances, it is not easy to precisely control the catalytic reaction of heterogeneous catalysis processes. The complexity and resolution of 3D-printed structures are restricted according to the specific limitations of the 3D printing methods. The most-reported method for shaping heterogeneous catalysts is Direct Ink Writing, which is an extrusion-based technique employed for fabricating 3D objects by extruding the ink of powders through a nozzle (generally, scaled at hundreds of micrometers). However, complex architectures are difficult to produce using this technique.

3D printing based on stereolithography enables the fabrication of 3D objects with a high complexity and resolution. This approach recently led to the formation of ceramic objects, such as those composed of alumina, zirconia, silica, and barium titanate [36]. The ink formulations mainly had a photo-polymerizable composition with dispersed ceramic particles. Very recently, SiOC, SiC, $Si_3N_4$, and $SiO_2$ structures with a high resolution were prepared with inks without particles, while the inorganic material was formed from a preceramic polymer or precursors of sol-gel reactions [7,8]. The feature dimensions can be tens of microns and even in the submicron range. For example, 3D photonic crystal structures with 220 nm diameter rods [37] and YAG miniature light emitters [38] were fabricated by Two-Photon Polymerization (TPP), which is a subcategory of 3D printing based on stereolithography. Therefore, with the help of this approach for 3D shapes with a much higher complexity and resolution, the possibilities of using 3D printing for controlling chemical reactions could be further expanded. This work introduces a general concept for fabricating 3D catalytic reactors by an approach based on stereolithography, with a solution composed of functional inorganic salts and photo-curable organics, dissolved in solvents, without any particles. This strategy provides a much higher resolution than existing approaches, and unlike reports on 3D printing with silicon-based pre-ceramic polymers or sol-gel, this new approach is very simple, because the precursors of target materials are simply dissolved salts. Moreover, this approach provides the possibility of producing multifunctional 3D architectures, for which several materials play a role as catalytic species by using several inorganic salts in the inks, while the geometries are defined by the intrinsic features of 3D printing (e.g., a sandwich honeycomb architecture of catalysis devices with heterogeneous pores where the ingredients of each layer are different and designed for the sequential catalytic reaction). Therefore, this approach provides a manufacturing method for fabricating high-complexity and -resolution multifunctional 3D reactors with simultaneous control of the chemical activity.

Herein, we demonstrate the feasibility of 3D printing catalysts with high-complexity structural features. $PtO_2$-$WO_3$ heterogeneous catalysts were fabricated with a solute-based ink, via digital light processing (DLP), which is a subcategory of stereolithography-based 3D printing, followed by pyrolysis of the object to obtain the functional 3D catalyst. The catalytic properties of 3D-printed architectures were tested in the hydrogenation of alkynes and nitrobenzene. The 3D-printed catalysts exhibited excellent reactivity in these catalytic transformations. Furthermore, to prove the concept of multifunctional reactors that enables sequential catalysis, a tungsten oxide-based tube consisting of three orderly sections containing platinum, rhodium, and palladium was 3D printed.

## 2. Results and Discussion

### 2.1. Preparation of 3D PtO$_2$-WO$_3$ Catalysts

The ink was prepared by dissolving the precursor of the catalyst (tungsten and platinum salts) and photo-curable organics consisting of acrylic acid as a monomer and polyethylene glycol diacrylate (PEGDA) as a crosslinker, in triple distilled water (TDW). Before printing, a 2,4,6-trimethylbenzoyl-diphenylphosphine oxide (TPO) photoinitiator was added. The printing process is schematically presented in Figure 1a, along with an example of a 3D-printed PtO$_2$-WO$_3$ catalyst at various stages of the fabrication process. The resulting translucent objects contained about 32% water and uncured acrylic acid (Figure 1(b-2) and (c-2)) that had to be removed by evaporation. After pyrolysis at a high temperature, a black-color catalyst with three-dimensional features was obtained (Figure 1(b-3) and (c-3)). The drying was performed by keeping the 3D object at room temperature for 3 days, while the pyrolysis was carried out by heating the objects at 450 °C for 60 min.

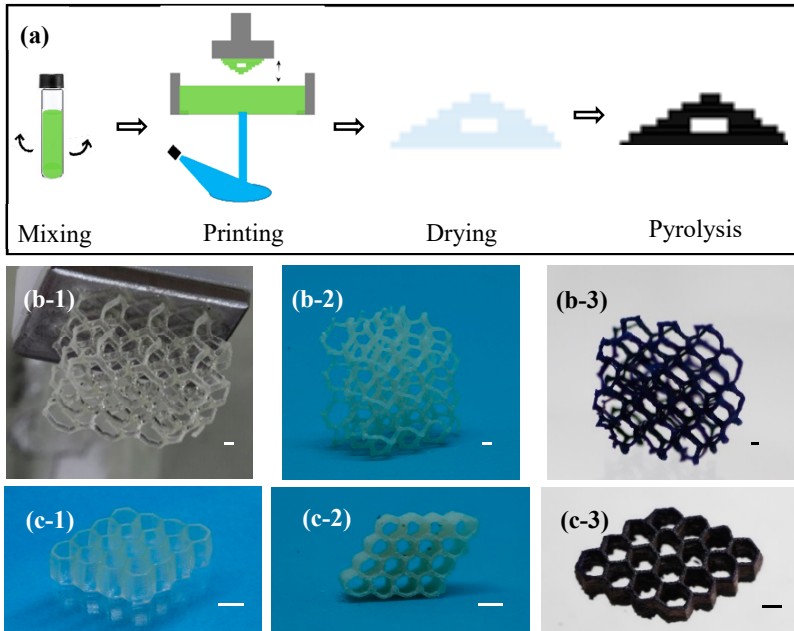

**Figure 1.** Schematic presentation of the printing process and examples of printed objects of catalysts obtained via 3D printing; (**a**) 3D printing schematic presentation; (**b-1**) and (**c-1**) present wet 3D objects; (**b-2**) and (**c-2**) display dried 3D objects; (**b-3**) and (**c-3**) show pyrolyzed 3D objects. The scale bars are 1 mm.

The precursors were initially dissolved in the ink, and gradually precipitated out during printing and drying, before finally existing in the form of particles trapped within the organic network formed during photo-polymerization. During the pyrolysis step, the precursors were transferred from salt into an oxide, with the simultaneous burning out of the organic materials. The decomposition of ammonium metatungstate hydrate had three stages (Figure S1): Losing absorbed water and crystal water below 180 °C; releasing ammonia gas between 180 and 380 °C; and gradually forming tungsten oxide above 380 °C. The weight losses in each stage was 2.7%, 4.65%, and 1%, respectively (Figure S1), in agreement with the literature [39,40]. As shown in Figure S1, the organic material without the salt completely degraded at temperatures above 450 °C. Meanwhile, potassium tetrachloroplatinate was converted into platinum oxide during heating [41]. Therefore, pyrolysis of the 3D object was almost completed at the same temperature, for which the weight loss was approximately 20%. The high-temperature treatment process resembles previous work [42–44] on the polyacrylamide-gel method [45] for making nano-powder, i.e., utilizing pyrolysis of the polyacrylamide xerogel containing inorganic salt to prepare powders of single elements, such as Al$_2$O$_3$ [46], ZrO$_2$ [47], or multiple-element

nanopowder, e.g., $2SiO_2$–$3Al_2O_3$ (mullite) [48] and $La_{0.85}Sr_{0.15}Ga_{0.85}Mg_{0.15}O_{2.85}$ [49]. The difference from the polyacrylamide gel method is that, here, the formed product retained its 3D structure and was not obtained as a powder. The main reason for keeping the 3D shape is that the degradation of polyacrylic acid-PEGDA gel in this work is more moderate than that of polyacrylamide gel; the complete decomposition temperature of the former is 450 °C, which is much lower than that of the polyacrylamide, with a value of 600 °C [42,43]. Additionally, the shrinkage of the 3D objects was low, at around 25%, since the salt was homogeneously dispersed within the objects and the salt concentration in this work was higher.

## 2.2. The Microstructure of the Printed $PtO_2$-$WO_3$ Catalyst

The microstructures and constituents of the pyrolyzed $PtO_2$-$WO_3$ catalysts are presented in Figure 2a,b. It is clear that the 3D structures retained their primary morphology compared to that obtained after printing and drying (as shown in Figure 1), although it was sintered and pyrolyzed at a high temperature (450 °C). Moreover, the layered feature of the 3D structures is very obvious, which stemmed from the layer-by-layer 3D printing process. In addition, the distribution of elements, including oxygen, carbon, tungsten, and platinum, was investigated by an Energy Dispersive Spectrometer (EDS) and is presented in Figure 2c. These elements were homogeneously dispersed in the structures. This is due to the major advantage of the presented approach, which utilizes a solution-based ink. The elements that are homogeneously mixed in the form of ions in the ink are trapped in the pores of the organic networks generated during the photopolymerization and are nailed in their position during the following process of sintering and pyrolysis. This resembles that of the polyacrylamide gel route for nanopowder [40,42], as aforementioned in Section 2.1. Interestingly, the particles of platinum oxide are nano-sized (Figure S2), which should result in a benefit for the catalytic properties of 3D structures.

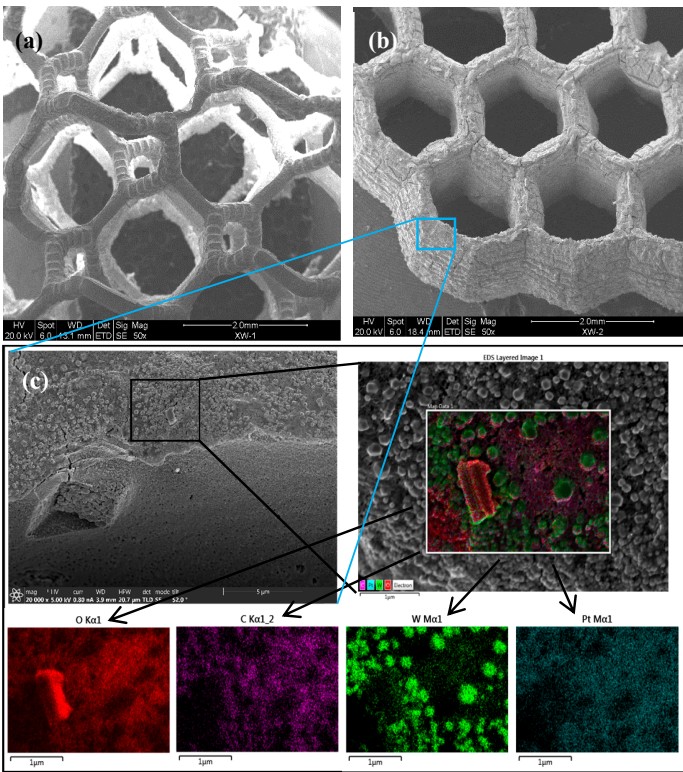

**Figure 2.** The microstructure and element distribution of the printed catalyst: (**a**) and (**b**) scanning electron microscopy (SEM) images; (**c**) element distribution (Energy Dispersive Spectrometer (EDS)).

### 2.3. Composition of the Obtained PtO$_2$-WO$_3$ Catalyst

Figure 3 shows the XRD patterns of the obtained catalyst, which indicates that it is composed of pure tungsten trioxide (WO$_3$). However, the WO$_3$ is composed of two phases—h-WO$_3$ (JCPDS 72-0677) and m-WO$_3$ (JCPDS 85-2460)—which are 15.1% and 84.9%, respectively, as calculated by the RIR method. The phase of WO$_3$ depends on its synthesis route. The phase of h-WO$_3$ is a hexagonal crystal, which is labile and generally synthesized by the hydrothermal method [50,51]. The phase of m-WO$_3$ is a monoclinic crystal [52], which is much more stable than that of h-WO$_3$ due to its synthesis route based on high-temperature calcination being prone to crystallization [53]. Moreover, the broadened diffraction peaks observed also indicate that the crystalline size of the sample is very small, in agreement with the nano-sized particles observed in Figure 2 and Figure S2. However, no platinum compound was detected in the XRD analysis, probably due to its very low content, which cannot be detected by this method.

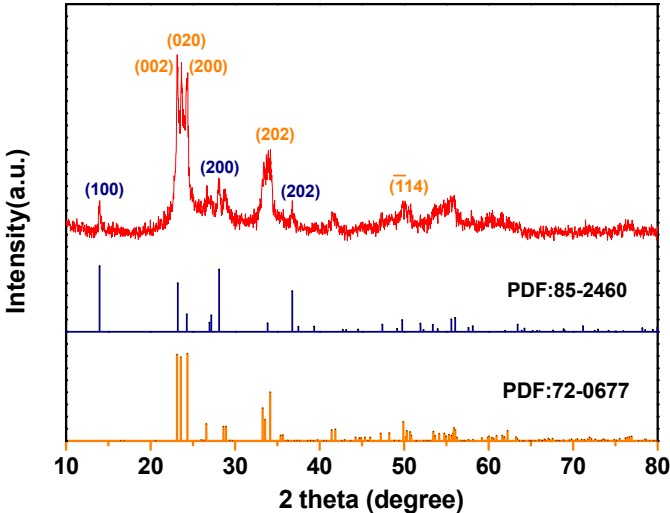

**Figure 3.** X-ray diffraction pattern of the pyrolyzed 3D object.

### 2.4. Properties of the Printed Catalyst

The obtained PtO$_2$-WO$_3$ 3D catalyst was tested in the semi-hydrogenation of alkynes as a model reaction. This catalytic transformation is of great importance in organic synthesis and industrial processes. It is considered a simple and straightforward method for preparing alkenes from readily-available alkynes. Alkenes are applied as starting materials in numerous significant organic transformations, such as polymerization, epoxidation, and hydroformylation. The catalytic semi-hydrogenation of alkynes was performed in 1,2-dimethoxyethane under 400 psi of hydrogen and the Pt:substrate ratio was 1:1330. The results of these reactions are presented in Table 1.

The 3D catalyst was very active in the hydrogenation of phenylacetylene as full conversion was obtained after 6 h (Table 1, entry 1). The selectivity toward styrene was 82%. The catalyst showed excellent reactivity when other derivatives of phenyacetylene with electron-withdrawing or electron-donating substituents were applied. Therefore, the hydrogenation of 1-ethynyl-4-fluorobenzene under the same conditions gave full conversion with a lower selectivity toward the alkene product—4-fluorostyrene (Table 1, entry 2). The substrates 1-ethynyl-4-methoxybenzene and 1-ethynyl-4-methylbenzene could be efficiently hydrogenated with 100% conversion (entries 3 and 4, Table 1, respectively). The selectivity in the case of 1-ethynyl-4-methoxybenzene substituted with a strong electron-denoting group was better than 1-ethynyl-4-methylbenzene, which has a moderate electron-donating group. However, there was no effect of the substituents on the selectivity of the hydrogenation, as 1-ethynyl-4-fluorobenzene with

a strong electron-withdrawing group and 1-ethynyl-4-methylbenzene with electron-donating groups exhibited similar selectivity values.

　　The hydrogenation of prop-1-yn-1-ylbenzene, which is an alkyne with an internal triple bond, proceeded with the conversion of 72% and selectivity of 83% toward the alkene products *cis* and *trans*-prop-1-en-1-ylbenzene (entry 5, Table 1). The decrease in reactivity of the catalyst in this case is perhaps due to steric effects. A decrease in the reactivity and selectivity of the 3D catalyst was also observed in the hydrogenation of aliphatic alkynes, such as 1-octyne (entry 6, Table 1).

**Table 1.** Semi-hydrogenation of alkynes [1].

| Entry | Alkyne | Catalyst | Conversion (%) [2] | Selectivity of Alkene (%) [2] |
|---|---|---|---|---|
| 1 | | $PtO_2$-$WO_3$ | 100 | 82 |
| 2 | | $PtO_2$-$WO_3$ | 100 | 72 |
| 3 | | $PtO_2$-$WO_3$ | 100 | 70 |
| 4 | | $PtO_2$-$WO_3$ | 100 | 82 |
| 5 [3] | | $PtO_2$-$WO_3$ | 72 | 83 |
| 6 | $n$-$C_6H_{13}$ | $PtO_2$-$WO_3$ | 75 | 67 |
| 7 | | Pt/C (10%) | 100 | 0 |
| 8 | | $PtO_2$ | 100 | 0 |

[1] Reaction conditions: alkynes (1.5–2.5 mmol), catalyst (1.1–2.1 μmol), 2 mL 1,2-dimethoxyethane, 400 psi $H_2$, 60 °C, 6 h; [2] determined by [1]H-NMR; [3] the ratio of *cis:trans* is 4:1.

　　Importantly, the printed catalyst showed excellent selectivity compared to commercially available platinum-based heterogeneous catalysts such as Adams catalyst, $PtO_2$, and Pt/C (10%). While the 3D catalyst could catalyze the hydrogenation of phenylacetylene with 82% selectivity, the catalysts Pt/C (10%) and $PtO_2$ were not selective and led to the formation of the non-desired product ethylbenzene (entries 7 and 8, Table 1, respectively). We believe that the selectivity of the $PtO_2$-$WO_3$ 3D catalyst in the hydrogenation of alkynes is probably related to electronic metal-support interactions and/or steric

effects caused by the WO$_3$ support around the active sites. It should be mentioned that the PtO$_2$ is most likely actually converted in the metal Pt, but due to its very low concentration, this could not be verified.

The 3D catalyst was also tested in the hydrogenation of another functional group—the nitro group. Therefore, the hydrogenation of nitrobenzene in dichloromethane under 400 psi hydrogen yielded 100% aniline after 6 h. However, the Pt/C (10%) and PtO$_2$ catalysts also showed the same reactivity in the hydrogenation of nitrobenzene.

### 2.5. WO$_3$-Based Catalyst with Multi-Catalytic Species

As aforementioned, we developed a simple approach for fabricating 3D catalysts with high structural complexity. Furthermore, this approach could be designed and branched out for performing sequential chemical reactions, i.e., designing a reactor containing multiple elements for catalytic reactions, e.g., a kit-catalyst with a special architecture. A tungsten oxide-based tube containing three catalysts—platinum, rhodium, and palladium—at separate locations within the tube (Figure 4), was printed via combining DLP printing with vat exchange, i.e., using vats with dissolved salts of tungsten Pt, Rh, or Pd, respectively. The different colors shown in Figure 4a,b confirm that the three-species catalyst was printed successfully. Figure 4 also shows that the 3D catalyst kept its shape while converting from the wet object to the pyrolyzed one, although some cracks were formed during its pyrolysis (Figure 4c). The three different colors shown in Figure 4a,b suggest that three elements were capped as our design and printed, which is also proven by the SEM observation (Figure S4). Interestingly, the microstructures of the three sections in the object are different from each other, as shown in Figure S4. These differences might be due to the fact that (1) the photopolymerization behavior is affected by the salts; (2) the diffusion of elements and crystallization behavior in wet gels is different for each precursor; and (3) the pyrolysis of each salt is different. This interesting result will be further studied in our future work.

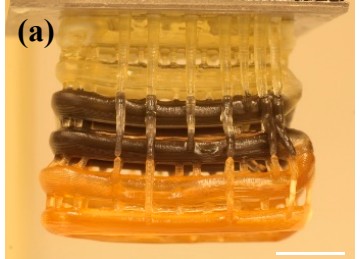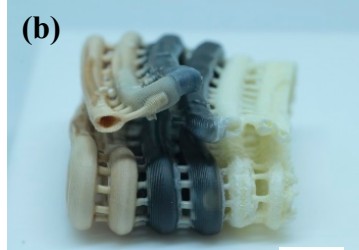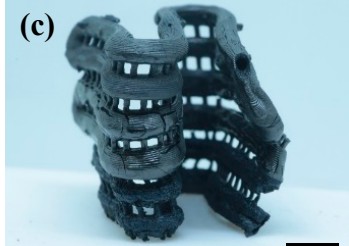

**Figure 4.** Images of the printed tube with three sections containing Pt, Rh, and Pd, successively: (**a**) Wet tube; (**b**) dried tube; and (**c**) pyrolyzed tube. The scale bars are 1 cm.

## 3. Materials and Methods

### 3.1. Materials

Ammonium metatungstate hydrate (H$_{40}$N$_{10}$O$_{41}$W$_{12}$•xH$_2$O, MW 2956.3) was purchased from Alfa Aesar (Heysham, UK). Potassium tetrachloroplatinate (Cl$_4$K$_2$Pt) was obtained from ACROS Organics (Belarus, Belgium). Acrylic acid was purchased from Sigma Aldrich (Merck, Germany). Polyethylene glycol diacrylate (PEGDA) was obtained from Sartomer-Arkema (Colombes Cedex, France). 2,4,6-trimethylbenzoyl-diphenylphosphine oxide (TPO) was obtained from BASF (Ludwigshafen, Germany). Triple distilled water (TDW) was used throughout all experiments. All alkynes and nitrobenzene were purchased from Aldrich and used without further purifications. Rhodium trichloride and palladium acetate were obtained from Strem Chemicals. All organic solvents were purchased from Biolab.

### 3.2. Ink Preparation

The ink for printing $PtO_2$-$WO_3$ catalysts was prepared by dissolving tungsten salt (36.37% w/w), platinum salt (0.34% w/w), acrylic acid (27.3% w/w), PEGDA (2.73% w/w), and TPO (0.55% w/w) in the TDW (32.71% w/w) with magnetic stirring. The weight ratio between acrylic acid used as the monomer and PEGDA used as the crosslinker was 10. The weight percentage of platinum in the $PtO_2$-$WO_3$ catalyst was 0.46%. The ink for printing a tungsten oxide-based reactor-tube contained three species of Pt, Rh, and Pd, which is almost the same as that for the $PtO_2$-$WO_3$ catalyst; only the salt contents of catalysts (Pt, Rh, and Pd) are different. The weight percentages of Pt, Rh, and Pd in each section of the tungsten oxide-based tube were 0.46%, 0.5%, and 0.5%, respectively.

### 3.3. Ink Printing

3D printing was performed by using a DLP 3D printer (Pico 2, Asiga, Alexandria, Australia) equipped with a UV-LED light source of 385 nm. The curing time of each layer, 200 µm thick, was 2−10 s. To print the tungsten oxide-based reactor-tube containing three species, the vat loading inks with Pt, Rh, and Pd were successively changed.

### 3.4. Drying and Pyrolysis Process

The obtained structures were dried in air at room temperature over approximately 3 days and subsequently decomposed in a tube furnace (Kejia Furnace, Zhengzhou, China) by heating to 450 °C for 60 min at the heating rate of 1 min/°C in air. The decomposition temperature of the dried 3D structure and heating rate was set according to the TG curve of dried 3D objects, tungsten salt, and purely organic material, as shown in Figure S1. The shrinkage was determined by measuring the dimensions of the printed structure after drying and of the final structure after heating at 450 °C, as shown in Figure S3.

### 3.5. Catalysis Reactions

The obtained $PtO_2$-$WO_3$ catalyst containing 1.1–2.1 µmol Pt and 1.5–2.5 mmol substrate in 2 mL solvent was transferred to a 15 mL glass-lined autoclave. The autoclave was sealed, purged with hydrogen (3X), and pressurized with hydrogen to 400 psi. The autoclave was heated at 60 °C for 6 h. Upon completion of the reaction, the pressure was carefully released and the conversion and selectivity were determined through [1]H NMR analysis of the reaction mixture.

### 3.6. Characterization

The crystalline phase of the final $PtO_2$-$WO_3$ catalyst was determined by an X-ray diffractometer D8 ADVANCE (Bruker AXS, Karlsruhe, Germany). Ultrahigh-resolution scanning electron microscopy (SEM) imaging and EDX were performed by Magellan XHR SEM field emission instruments (FEI, Hillsboro, OR, USA), without a conductive coating; ESEM imaging was accomplished by employing a Quanta 200 FEG (FEI, Hillsboro, OR, USA). [1]H-NMR spectra were recorded with a Bruker DRX-400 instrument.

## 4. Conclusions

A versatile approach was developed to print 3D catalysts with high structural complexity by the digital light processing of solutions combined with pyrolysis. The inks were clear solutions produced by simply mixing photocurable monomers, PI, and metallic salts, which is extremely beneficial for fabricating complex shapes. This 3D catalyst maintained its architecture, even after a series of post-treatment processes, including drying at ambient temperature and pyrolyzing under a high temperature (450 °C), which is due to its controlled shrinkage resulting from the high salt concentration in the inks. The 3D-obtained catalysts tested in the hydrogenation of alkynes and nitrobenzene showed excellent reactivity in these catalytic transformations.

A tungsten oxide-based tube that consisted of three successive sections containing platinum, rhodium, and palladium was successfully 3D printed, indicating that the approach reported in this paper is feasible for multifunctional reactors, e.g., kit-catalysts. The possibility of designing new catalytic materials with new structures and controlling their mechanical and chemical properties is feasible with 3D printing and this can enable facile tuning of the reactivity and selectivity of catalysts.

**Supplementary Materials:** The following are available online at http://www.mdpi.com/2073-4344/10/8/840/s1: Figure S1. Thermogravimetric analysis of tungsten salt, ammonium metatungstate hydrate, a dried 3D object, and a pure organic substance without salt, which were fabricated in the same route as that for the 3D object; Figure S2. The particles of platinum oxide in the obtained catalyst, which is nano-sized; Figure S3. SEM images of the pyrolyzed object; Figure S4. SEM imagines of the pyrolyzed tube. (a) and (b) the interface between sections with Pd and Rh; (c) the section with Rh; (d) the section with Rh; (e) the interface between sections with Pt and Rh; (f) the section with Pt, which is similar to that shown in Figure 2.

**Author Contributions:** Conceptualization, X.W., R.A.-R., and S.M.; methodology, X.W., W.G., R.A.-R., and S.M.; formal analysis, R.A.-R. and S.M.; investigation, X.W., W.G., and R.A.-R.; resources, R.A.-R. and S.M.; data curation, R.A.-R. and S.M.; writing—original draft preparation, X.W. and S.M.; writing—review and editing, X.W., R.A.-R., and S.M.; supervision, R.A.-R. and S.M.; project administration, S.M.; funding acquisition, R.A.-R. and S.M. All authors have read and agreed to the published version of the manuscript.

**Funding:** This research was funded by joint funding of The Hebrew University (HU) and Central South University (CSU), HUCNN-CSU-2019, and by the Israel Ministry of Science and Technology.

**Acknowledgments:** We thank the Israel Ministry of Science and Technology for the partial financial support. X.W. gives thanks for the financial support from CSC, the Chinese Scholarship Council of the Ministry of Education, China. We also thank Sartomer-Arkema and their agent in Israel, Eltra, for providing the PEGDA sample.

**Conflicts of Interest:** The authors declare no conflict of interest.

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
