# Peer review of "High-Complexity WO3-Based Catalyst with Multi-Catalytic Species via 3D Printing"

_catalysts, doi:10.3390/catal10080840_

Round 1

Reviewer 1 Report

Congratulations to the authors for their good work on this manuscript. The following comments help you improve the quality of your work prior to publication:

  • in the introduction, please add a paragraph on different applications of additive manufacturing, especially complex parts which uses more recent publications and update the source of your paper. you can mention fabricating complex parts, biomedical devices, coatings, industry 4.0, etc. the following articles are strongly recommended to add this paragraph.

https://link.springer.com/article/10.1007/s10853-019-03375-1 (biomedical devices)

https://www.mdpi.com/2075-4701/9/8/811 (metal extrusion)

https://www.mdpi.com/2076-3417/9/18/3865 (industry 4.0)

https://www.sciencedirect.com/science/article/pii/S0955221918306782 (3D printing of ceramics)

  • in figure 2, please indicate where is the location of the image "c" in the structure.
  • in figure 4, what is the "imagines" of the printed ...?
  • in the processing, what how did you evaporate the solvents of the ink? is it safe to use it widely?
  • it would be better if you could do more characterization tests.
  • please make the conclusion in bullet points to make it easier for readers to catch the summary of your work in a glance.

Good luck!

Author Response

Response to Reviewer 1 comments

(ID: catalysts-862810)

Point 1: In the introduction, please add a paragraph on different applications of additive manufacturing, especially complex parts which uses more recent publications and update the source of your paper. you can mention fabricating complex parts, biomedical devices, coatings, industry 4.0, etc. the following articles are strongly recommended to add this paragraph. https://link.springer.com/article/10.1007/s10853-019-03375-1 (biomedical devices); https://www.mdpi.com/2075-4701/9/8/811 (metal extrusion); https://www.mdpi.com/2076-3417/9/18/3865 (industry 4.0); https://www.sciencedirect.com/science/article/pii/S0955221918306782 (3D printing of ceramics)

Response 1: Thanks for the suggestion. The background of 3D printing applications is updated. The recommended articles have been cited in the revised manuscript. (Lines 33-37)

Point 2: In figure 2, please indicate where is the location of the image "c" in the structure.

Response 2:Many thanks. The location of the image "C" is at a place in the image "b", has added in Fig.2. (Lines 146-149)

Point 3: In figure 4, what is the "imagines" of the printed ...?

Response 3:It is a typo, it should be the "images" . We corrected it.

Point 4: In the processing, what how did you evaporate the solvents of the ink? is it safe to use it widely?

Response 4:In the processing, the solvents were evaporated in a hood. This approach is safe and prone to be used widely.

Point 5: It would be better if you could do more characterization tests.

Response 5:Thanks for your constructive suggestion. The purpose of the present manuscript is to provide a proof of concept for our approach, and we plan to do more comprehensive research on this work in the future.

Point 6: please make the conclusion in bullet points to make it easier for readers to catch the summary of your work in a glance.

Response 6: According to this suggestion, the conclusion section was divided into 2 paragraphs. (Lines 272-284)

Reviewer 2 Report

see attached pdf

Author Response

Response to Reviewer 2 comments

 (ID: catalysts-862810)

Point 1: Line 198: “… the <space> nitro …”

Response 1:Thanks for noticing. It is corrected (Line 204)

Point 2: Refs. [11] – [13]: inclusive pagination suggests all three of these articles are a single page? Clarify.

Response 2:We appreciate the reviewer’s careful reading. Reference [11] has 14 pages, but the page number in this journal is 4. Reference [12] has 14 pages, but the page number in this journal is 41. The page numbers of Refs. [11] & [12] were changed to be one number. (Line 323 and 325) The page number of reference [13] is wrong and has been changed from "42-42" to "42-43". We apologize for this mistake. (Line 330)

Point 3: Refs. [31] & [33]: presumably the numbers shown as page numbers should be single article reference numbers?

Response 3:Thanks for the careful checking. Reference [31] has 8 pages, but the page number in this journal is 1900604. Reference [33] has 8 pages, but the page number in this journal is 2001675. The page numbers of Refs. [31] & [33] were changed to be one number. (Line 366 and 370)

Point 4: Ref. [40], line 383: meaning of “[J]” ?

Response 4:We made a mistake. The "[J]" was deleted according to the format of Catalysts.

Reviewer 3 Report

In this manuscript, the authors presented an article entitled “High complexity WO3-based catalyst with multi-catalytic species via 3D printing”. The work describes a novel approach in 3D printing to fabricate catalysts with complex architectures using digital light processing. The authors prepared PtO2-WO3 heterogeneous catalyst via this method and, characterized. Then, they tested the utility of this catalyst towards selective hydrogenation of Alkynes and, as well as for the reduction of nitrobenzene. Also, the authors have prepared another WO3- based catalyst tube with platinum, rhodium, and palladium, via this 3D printing approach. Overall, the manuscript was written well with appropriate references, however, found some typographical errors.

There are some important points to be addressed.

1. The semi-hydrogenation of alkynes is an excellent method of use. However, it would be good if authors can provide some rationale for this ‘poisoned’ reactivity of the PtO2.

2. On page-8, line-55, the catalysis reaction protocol says PtO2-WO3, 3D printed catalyst with 5.5 mmol Pt, was used for 1.5-2.5 mmol substrate. If this is correct, that is a whole lot of Pt, which makes this approach of no synthetic use.

3. Line-265 & 266, it was described GC was used to track the reaction progress and selectivity of the hydrogenation reactions. However, in the table-1, it was mentioned 1H NMR was used. It should be corrected.

Author Response

Response to Reviewer 3 comments

 (ID: catalysts-862810)

Point 1: The semi-hydrogenation of alkynes is an excellent method of use. However, it would be good if authors can provide some rationale for this ‘poisoned’ reactivity of the PtO2.

Response 1:We thank the reviewer for his valuable comment. We added a possible reason for the selectivity of our catalyst. (Line 199-201)

Point 2: On page-8, line-55, the catalysis reaction protocol says PtO2-WO3, 3D printed catalyst with 5.5 mmol Pt, was used for 1.5-2.5 mmol substrate. If this is correct, that is a whole lot of Pt, which makes this approach of no synthetic use.

Response 2:The reviewer is right. This mistake was corrected (Line 187 and 261). The correct amount of the catalyst used in the catalytic hydrogenation was 1.1-2.1 µmol.

Point 3:  Line-265 & 266, it was described GC was used to track the reaction progress and selectivity of the hydrogenation reactions. However, in the table-1, it was mentioned 1H NMR was used. It should be corrected.

Response 3:We thank the reviewer for his comment. This was corrected and the description of the GC method was deleted. (Line 270)

Round 2

Reviewer 1 Report

Thank you for your modifications.

On line 36, is it correct to have term "on cite" or you meant "on site"?

Thanks and good luck with your research.